# Green Dentistry: State of the Art and Possible Development Proposals

**DOI:** 10.3390/dj13010038

**Published:** 2025-01-16

**Authors:** Stefano Speroni, Elisabetta Polizzi

**Affiliations:** 1Department of Dentistry, Dental School, IRCCS San Raffaele Hospital, Vita-Salute San Raffaele University, 20132 Milan, Italy; speroni.stefano@hsr.it; 2Chair Center for Oral Hygiene and Prevention, Department of Dentistry, Dental School, IRCCS San Raffaele Hospital, Vita-Salute San Raffaele University, 20132 Milan, Italy

**Keywords:** dentistry, operative, waste management, dental waste, environmental health, dental materials

## Abstract

**Objectives**: The objective of this narrative literature review was to highlight all dental procedures attributable to sectoral waste and to consider possible alternatives in line with the concept of sustainable development. **Methods**: An extensive search of electronic databases, including the Cochrane Oral Health Group Specialized Register, Cochrane Central Register of Controlled Trials (CENTRAL), Web of Science, PubMed, EMBASE, and Google Scholar. Search words included ‘Green Dentistry’, ‘Dental Pollution’, ‘Pollutants and Dentistry’, ‘Disinfectants and Dentistry’, and ‘High-tech Dentistry’. All of them allowed an assessment of the impact of dental practice on the external environment, and new frontiers currently applied or possibly applicable for green dentistry were included in the study. Non-full-text papers, animal studies, studies in languages other than English, and studies not related to the topic under consideration were excluded. **Results**: According to the inclusion criteria, 76 papers were selected for the study. The topics analyzed were the impact of dental practice on the outdoor environment, currently applied and potentially applicable principles of green dentistry, and the ‘Four Rs’ model (Rethink, Reduce, Reuse, and Recycle). **Conclusions**: With the limitations of the present study, the concept of green dentistry could be applicable provided that the measures already taken to reduce indoor and outdoor risk factors are continued and improved.

## 1. Introduction

Human health is defined by the World Health Organization as “a state of complete physical, mental, and social well-being” rather than merely “the absence of disease or infirmity”, and it is closely linked to the environments in which social communities are established and their habits [1,2]. Consequently, without a healthy environment, public health, economic stability, and social stability can only be partially achieved [3]. According to conducted studies, pollutants, carcinogens, and pathogenic microorganisms released into the environment due to human activities or natural disasters significantly affect air quality and water and food supply, and thus the quality of life of communities. Therefore, considering what has been said, sustainability in every aspect of our lives is fundamental [4,5]. Sustainable development is a multidimensional concept aimed at meeting the needs of present generations without compromising the ability of future generations to meet their own needs. It encompasses three interrelated pillars: economic, environmental, and social sustainability. Economic sustainability emphasizes growth models that are inclusive, equitable, and maintain resources for long-term prosperity. Environmental sustainability aims to conserve biodiversity, reduce pollution, and ensure the responsible use of natural resources. Social sustainability focuses on creating systems that improve living standards. ensure fair access to resources, and promote equity and justice within communities [6,7].

In recent decades, development models have evolved to incorporate principles of sustainability, transitioning from resource-intensive, linear approaches to circular models that prioritize resource efficiency, waste reduction, and ecosystem health. Key characteristics of these models include renewable energy adoption, eco-design principles, and the integration of life-cycle assessments into planning and decision-making processes. Such frameworks align with the global agenda for sustainable development, as outlined in international agreements like the United Nations Sustainable Development Goals (SDGs) [8].

The principles of sustainable development are particularly relevant in the healthcare and dental sectors, where procedures generate significant amounts of waste and consume considerable resources. The dental industry, in particular, contributes to sectoral waste through the use of disposable materials, packaging, and chemical substances that pose challenges for waste management and environmental health. Addressing these issues involves rethinking the materials and methods used in dental practices to reduce ecological footprints and promote sustainability. Initiatives such as the adoption of biodegradable materials, recycling programs, and energy-efficient technologies are steps toward aligning dental procedures with sustainable development goals. In this scenario, the objective of this narrative literature review was to highlight all dental procedures attributable to sectoral waste and to consider possible alternatives in line with the concept of sustainable development.

## 2. Materials and Methods

From September 2023 to September 2024, an extensive search of electronic databases, including the Cochrane Oral Health Group Specialized Register, Cochrane Central Register of Controlled Trials (CENTRAL), Web of Science, PubMed, EMBASE, and Google Scholar, was performed. The SANRA scale was applied to assess study validity (https://rossisanusi.wordpress.com/wp-content/uploads/2022/03/sanra.pdf; access date: 4 September 2024).

Search words included ‘Green Dentistry’, ‘Dental Pollution’, ‘Pollutants and Dentistry’, ‘Disinfectants and Dentistry’, and ‘High-tech Dentistry’. All of them allowed an assessment of the impact of dental practice on the external environment, and new frontiers currently applied or possibly applicable to green dentistry were included in the study. Non-full-text papers, animal studies, studies in languages other than English, and studies not related to the topic under consideration were excluded.

The table below summarizes the criteria used for study selection (Table 1).

The decision-making process for study selection is illustrated in the following flow chart:Identification: The database search yielded a total of 1230 studies. After removing duplicates, 890 unique studies remained.Screening: Titles and abstracts were screened, resulting in the exclusion of 720 irrelevant studies.Eligibility: The full texts of 170 studies were assessed against the inclusion and exclusion criteria.Inclusion: A final selection of 76 studies was included for qualitative analysis.

The included studies analyzed the following key characteristics:Pollutants in Dentistry: Emissions, waste management, and use of non-biodegradable materials.Eco-Friendly Alternatives: The adoption of biodegradable materials, non-toxic disinfectants, and sustainable packaging.Energy Efficiency: The integration of energy-saving devices and technologies in dental practices.High-Tech Dentistry: Digital tools and systems reducing material waste and carbon footprints.Compliance and Awareness: Levels of compliance with environmental regulations and awareness among dental professionals about green practices.

## 3. Results

According to the inclusion and exclusion criteria, 76 studies were considered.

Impact of dental practice on the outdoor environment

Indoor air quality in the dental practice is a factor that affects the health of dental workers and patients. According to the article ‘What’s the air in dental practices’ published in the journal ‘Dentistry 33’ in May 2020, indoor air pollution can be up to 10 times worse than outdoor air pollution. Indeed, enclosed spaces allow potential pollutants to accumulate [9]. This is the reason for the recommendation to favor natural ventilation, which cannot, however, be considered sufficient: air pollution is, in fact, one of the main challenges of our time [10]. Indoor pollutants are divided into:○Chemical: mercury, methacrylate, heavy metals, solvents and disinfection chemicals;○Biological: viruses, bacteria, fungi, molds;○Physical: noise pollution and radiation [11].

### 3.1. Chemical Pollutants

#### 3.1.1. Mercury

Mercury (Hg), which constitutes approximately 50% of amalgam composition, is a heavy metal that can exist in elemental form or in organic and inorganic compounds with varying levels of toxicity [12]. Mercury is ranked as the third-most toxic element to human health by the United States (US) Government Agency [13]. According to Pure Earth, an international organization known for addressing pollution in low- and middle-income countries, there is a health risk associated with mercury exposure affecting 19 million people [14]. In 2015, the toxic sites identification program identified over 450 sites worldwide where mercury exposure threatens population health [15]. The adverse effects of mercury are well-documented; therefore, it is crucial to prevent its release into the atmosphere because once inside, changes in pH, oxygen availability, and temperature can allow mercury to be utilized by bacteria capable of converting it into ‘organic methylmercury’, a potent neurotoxin [16]. Due to its structural properties, methylmercury accumulates in the fatty tissue of organisms such as fish and mammals, which are consumed by humans, resulting in bioaccumulation in food chains [17]. Vulnerable populations, such as children, the fetuses of pregnant women, hypersensitive individuals, and those with renal insufficiency, are particularly susceptible to the neurotoxic effects of methylmercury [18]. The main symptoms of intoxication include chronic fatigue, frequent headaches or migraines, marked weight changes, excessive thirst, anemia, and persistent difficulty falling asleep [19]. Severe intoxication can affect various bodily systems, including the respiratory, nervous, reproductive, urinary, cardiac, musculoskeletal, auditory, olfactory, visual, digestive, and immune systems [20]. Diseases related to the toxicity of mercury include fibromyalgia, rheumatoid arthritis, multiple sclerosis, Parkinson’s disease, Alzheimer’s disease, Crohn’s disease, ADHD, and autism [21,22]. The severity of symptoms depends on several patient predisposing factors: the number of mercury fillings present, the time elapsed since operations, damage caused by chewing, occlusal problems, and bruxism [23]. The increasing control of mercury pollution and the protection of human health and the environment from its adverse effects have become political objectives on both a global and national scale [24]. In dentistry, mercury, like lead, is a component found in varying percentages within dental amalgam [25]. This restorative material, if not handled and disposed of correctly, can pose a serious threat to the biosphere and patient health, as its particles are known to be neurotoxic, nephrotoxic, and bio-accumulative [26].

In recent decades, its use has been drastically reduced, both for aesthetic reasons and for the aforementioned health and environmental concerns. With the Interministerial Decree of 11 November 2020, the National Plan for the Elimination of Dental Amalgam Use was adopted, in compliance with Regulation (EU) 2017/852 on mercury, which, in Article 10, paragraph 3, provides that each member state shall define a national plan concerning measures it intends to take to gradually eliminate its use. The plan, officially published on the Ministry of Health’s website on 23 February 2021, describes the actions necessary to achieve the progressive discontinuation of its use in the dental sector, with the aim of achieving complete elimination through non-coercive measures by 31 December 2024. The realization of this objective will be based primarily on a widespread information and training campaign targeting sector operators involved in the dental supply chain (manufacturers, distributors, professional associations, universities, and dental scientific societies) [27].

#### 3.1.2. Methacrylate

Over the past seventy years, polymethyl methacrylate (PMMA) acrylic resins have been among the most widely used materials for the fabrication of prosthetic and orthodontic devices [28]. PMMA is a volatile substance with excellent solvent properties. It is readily absorbed by our bodies, and is irritating to the skin, mucous membranes, and respiratory tract. The greatest risk is for operators who should handle it with appropriate protections, which, while recommended in safety data sheets, render the material’s manipulation impractical in normal laboratory protocols. Consequently, the operator is unable to work with the chemical material without risking harm to their health [29].

Regarding the use of devices in the oral cavity, it is now widely accepted that polymerized matrices of resinous material can release monomers responsible for cytotoxic reactions. Currently, from a technical and scientific standpoint, the issue of residual monomer release has not yet been resolved and remains an integral part of PMMA-related issues [30].

In a study published in the Journal of Oral Rehabilitation (2004), the cytotoxic effects of liquids from three different dental resins for in situ relining and their main components methyl methacrylate (MMA), isobutyl methacrylate (IBMA), and 1,6-hexanediol dimethacrylate, were highlighted. The results clearly show that all tested materials possess a certain level of cytotoxicity. Moreover, the release period can last for several years. The greatest harm to patients is undoubtedly associated with the direct use of these materials within the oral cavity. In fact, some in vitro studies report that the majority of treated cells died due to necrosis, while a small percentage died due to apoptosis. In conclusion, the results demonstrate that these dental polymer liquids and their main monomers cause cytotoxic reactions. The direct relining procedure involving the polymerization of these materials in situ should be used with great caution [31].

In a more recent study published in 2020 by Juráňová et al., the cytotoxicity of cross-linked methacrylate-based biopolymers through in situ photopolymerization was mainly attributed to residual methacrylate monomers released due to incomplete polymerization. They can irritate adjacent tissue and be released into the bloodstream, reaching all tissues. An increase in the production of reactive oxygen species was also observed following cell exposure to methacrylates. Reactive oxygen species can participate in genotoxic or pro-apoptotic effects induced by methacrylate and cell cycle arrest through the induction of corresponding molecular pathways in cells. A deeper understanding of the biological mechanisms and effects of widely used methacrylates in various bio-applications may allow for a better estimation of potential risks and, therefore, the selection of a more appropriate polymer material composition to eliminate potentially harmful substances such as triethylene glycol dimethacrylate [32].

From a Norwegian study in 2018 (Airborne exposure to gaseous organic and particle-associated substances in dental resin-based materials during restoration procedures, Department of Clinical Dentistry, UiT—The Arctic University of Norway, Troms), it emerges that procedures such as finishing resin prostheses or composite restorations increase the air concentration of semi-volatile and volatile methacrylates (HEMA, MMA, dimethacrylate, TEGDMA, BISGMA) and other microparticles and solvents. In the conclusions, the authors cite studies where exposure to these pollutants has been associated with more severe and frequent cases of allergic contact dermatitis, respiratory hypersensitivity, and respiratory diseases in general. Dentists, moreover, could be at a higher risk than the general population of developing idiopathic pulmonary fibrosis, and airborne exposure to chemicals and particulate matter in a professional environment is believed to play a key role [33].

#### 3.1.3. Heavy Metals

The danger of heavy metals such as copper, zinc, lead, and cadmium stems from their ability to bioaccumulate, meaning that they deposit in tissues [34]. The simultaneous presence of multiple metals within the oral cavity results in a mechanism called the “galvanic effect”, which produces ions (electric currents), which are often responsible for altering the pH of the mouth, facilitating the onset of neuralgia, migraines, headaches, and the proliferation of differentiated microbial strains. This combined mechanism (the lowering of the salivary pH value and ion migration) induces greater susceptibility to fungal infections (Candida Albicans) and parasitic infections of the oral cavity, adenoids, and tonsils (recurrent otitis, submandibular adenitis) [35].

Toxic metals released from the corrosion of dental materials, which the body cannot physiologically eliminate due to excessive accumulation, deposit in specific target organs for each metal. This is the case with mercury (Hg), which localizes in the central nervous system, causing immediate damage attributable to disturbing the electrical function of nerve cells and long-term damage affecting myelin tissues. In addition to mercury, the most used metals in dentistry include silver (Ag), copper (Cu), tin (Sn), zinc (Zn), cadmium (Cd), nickel (Ni), chromium (Cr), beryllium (Be), lead (Pb), palladium (Pd), platinum (Pt), gold (Au), and titanium (Ti), which are all present in combinations of various dental alloys.

Symptoms of toxicity can be local and systemic. Locally, manifestations may include tongue burning, metallic taste, dysphagia, bluish discoloration of the oral mucosa near the metal alloy (tattooing), facial and trigeminal neuralgia, pulpitis, and periodontitis [36].

Systemically, metallic ions released from alloy corrosion, through thermal and electrophoretic effects, reach the gastrointestinal tract and, through the mucosa, enter the bloodstream and lymphatic system, potentially localizing in organs far from the oral cavity, generating a wide range of generic symptoms such as intestinal dysbiosis, alteration of the heart rhythm, anxiety, chronic fatigue, premature aging, immunosuppression, and specific symptoms of organ pathology. Conditions such as dermatitis and eczema, psoriasis, and degenerative skin diseases with unknown etiology may be attributed to the role that heavy metals play, due to the significant increase in their usage [37].

#### 3.1.4. Solvents

Among the solvents commonly used in dentistry, specifically in the field of endodontics (which deals with the internal tissues of the tooth, related pathologies, and treatments), two main categories are prominent: hydrocarbon solvents, including xylene and chloroform, and solvents of natural origin, such as eucalyptol and limonene-based mixtures (e.g., GPR-OGNA and BIO-ORANGE) [38].

In dental practice, the use of solvents is common in various procedures, including endodontics, which focuses on the treatment of the internal tissues of the tooth. However, it is essential to consider the toxic effects of such solvents to ensure the safety of both patients and healthcare personnel.

Xylene: This hydrocarbon solvent is widely used for gutta-percha removal during endodontic therapy. However, prolonged exposure to xylene vapors or inhalation can cause respiratory tract irritation, headaches, dizziness, nausea, and in extreme cases, damage to the central nervous system [39].Chloroform: Although it was used in the past as a solvent in endodontics, chloroform is now of increasing concern due to its toxic effects and volatility. The inhalation of chloroform vapors can cause respiratory tract irritation, kidney and liver damage, and in extreme cases, damage to the central nervous system and even carcinogenic effects [40].Eucalyptol: This naturally derived solvent is often used for its antimicrobial and analgesic properties in dentistry. However, excessive use or prolonged exposure to eucalyptol can cause mucosal irritation, skin allergies, and irritation of the eyes and respiratory tract [41].Limonene: Found in mixtures such as GPR-OGNA and BIO-ORANGE, limonene is a solvent derived from citrus fruits. Although considered relatively safe, the prolonged exposure or inhalation of limonene vapors can cause respiratory and skin irritation, as well as allergic reactions in some sensitive individuals [42].

#### 3.1.5. Disinfectants

Disinfectants encompass a broad group of substances aimed at destroying, eliminating, reducing, or preventing the action of bacteria, viruses, spores, and, in some cases, algae or other microorganisms [43].

In the field of dentistry, the clinical uses of disinfectants can be summarized as follows:○The cleaning and disinfection of dental impressions, made of various materials (addition and condensation silicones, hydrocolloids, polyethylene gums, alginates, polyethers, and polysulfides), which pose a high infectious risk due to the impossibility of sterilizing them in an autoclave;○The disinfection of surfaces (instruments, equipment, and non-invasive devices);○The disinfection of the aspiration system.

Chemical disinfectants can be classified into three categories based on their effectiveness against vegetative bacteria, tubercular bacilli, fungal spores, and viruses, namely high-level disinfectants, medium-level disinfectants, and low-level disinfectants [44].

High-level disinfectants: High-level disinfectants are capable of inactivating bacterial spores and all other microbial forms and can be listed as follows:
○Paracetic acid < 1%; the concentration in question is not harmful to human health [45];○Glutaraldehyde. In the dental field, the concentration used is 2%. Glutaraldehydes are bactericidal, virucidal, fungicidal, sporicidal, and parasiticidal, and can be highly toxic. In concentrations higher than 50% (the case in which they are used in medical environments to prepare diluted solutions), it is considered a toxic substance for inhalation and ingestion and is corrosive upon skin contact. Exposure mainly occurs through inhalation and skin contact during the handling and disinfection treatment of medical materials. It spreads into the environment through evaporation, primarily affecting the eyes and respiratory tract. Considering the widespread use of this substance by dentists, where possible, reducing its use or substituting it with others with comparable activity is advisable [46];○Ortho-phthalaldehyde 0.55%. At this concentration, it kills bacteria, viruses, fungi, and mycobacteria. If used for a longer time, it functions as a sporicidal agent. This product is not considered hazardous; however, it is advisable to handle it in a ventilated environment and wear personal protective equipment to avoid the irritative effects induced by it [47];○Sodium hypochlorite. The toxic characteristics of sodium hypochlorite solutions depend on their concentration. Those with the highest concentration (over 10% active chlorine) have a highly irritating effect. The ingestion of sodium hypochlorite can cause irritation or caustic injury in the mouth, throat, esophagus, or stomach. Sodium hypochlorite can be harmful only if mixed with other substances, resulting in the release of toxic gases. Often, this substance is erroneously used with acids—a practice that can produce chlorine gas, causing poisoning with typical symptoms such as a cough, dizziness, nausea, respiratory disturbances, the severe irritation/inflammation of mucous membranes, and conjunctivitis. The consequences can be respiratory failure or pulmonary edema [48].

Medium-level disinfectants: Medium-level disinfectants, namely formaldehyde, chlorine compounds, iodophors, alcohols, and phenolic compounds, destroy microbes such as tubercular bacilli but do not inactivate spores.

○Formaldehyde. At concentrations ranging from 0.1 to 3 ppm, it causes an irritation of the airways; for concentrations exceeding 10 ppm, a sense of suffocation is observed, while exposures above 20 ppm can lead to pulmonary edema. According to some data reported by ANDI, the first reports on the carcinogenic potential of formaldehyde date back to 1979, following studies conducted on experimental animals. Regarding humans, the carcinogenic action has been examined by the IARC, concluding that the evidence is considered limited (some of the studies examined have reported moderately significant increases in the frequency of malignant neoplasms of the oral cavity, nasal fossae, paranasal sinuses, and lungs, while others have not found such increases) [49];○Iodophors. Among these, povidone iodine is an antimicrobial and antiseptic active ingredient used for the disinfection and cleansing of skin and some types of mucous membranes. Its detergent solutions are used for surgical hand washing and also for disinfecting surfaces or objects in high-risk healthcare areas [50];○Phenolic derivatives. Phenolic derivatives have a broad biocidal spectrum except for spores and some viruses such as hepatitis A. They are used for disinfecting walls and floors and for pre-decontaminating surgical instruments and surgical hand washing. The polyphenol mixtures currently used are biodegradable; however, their discharge can only occur through the sewage system if the phenol concentration is <1 mg (Legislative Decree 152/2006) [51,52].

Low-level disinfectants:○Quaternary ammonium derivatives. They have a limited antimicrobial spectrum and are not volatile. They are inactivated by organic material; therefore, they are not suitable for disinfecting materials contaminated by biological fluids [53];○Benzalkonium chloride (BC) 0.25%. It is a quaternary ammonium chloride salt (QA) with antibacterial, antiseptic, detergent, and surfactant action. QA compounds are of low toxicity, but prolonged contact can irritate body tissues [54];○Chlorhexidine (CHX). It is an antiseptic disinfectant with a broad spectrum of action against Gram-positive and Gram-negative bacteria, some species of pseudomonas, and fungi; it inhibits mycobacteria, including the one responsible for tuberculosis, and is active against some viruses. Depending on its concentration, it may cause transient side effects, the most common being taste alteration and dental discoloration [55]. In this regard, some chlorhexidine-based products are specifically formulated to prevent or limit the degree of tooth discoloration and are low in toxicity. Dentists and dental hygienists will know how to suggest appropriate administration methods and concentrations for each individual patient [56];○Alcohols. The most effective bactericidal action is achieved in the presence of water (dilution to 70%). Alcohols are flammable; therefore, storage should be away from open flames. They can irritate tissues. The main alcohols used in dentistry are isopropyl alcohol, ethyl alcohol, and triclosan. The latter is known for its toxicity and possible sequelae associated with overexposure: muscular and cardiac problems, and skin, intestinal, and endocrine disorders [57]. Furthermore, a study in the “Proceedings of the National Academy of Sciences” highlights how prolonged exposure to triclosan can lead to the development of tumors, especially in the liver [58].

### 3.2. Physical Pollutants

Physical pollution is characterized by a very wide range of agents that can be harmful to human health if they exceed certain threshold levels.

The main physical pollutants are:○Noise pollution;○Ionizing radiation;○Electromagnetic fields.

#### 3.2.1. Noise Pollution

According to an article in the Italian Dental Journal, noise pollution, especially in the case of continuous exposure, is a source of nervousness and reduced concentration. In addition, excessively loud sounds can affect the quality of the specialist’s work [59].

Myers et al. found that dentists are significantly more affected by tinnitus than the general population, although they do not differ from the general population in terms of prevalence of hearing impairment per age group [60].

On the other hand, research carried out in South America by Lopes et al., in a study on the high-frequency hearing thresholds of dental professionals, showed that dentists have a higher risk of losing their hearing in high frequencies (between 9000 and 16,000 Hz), which are normally not included in phonometric examinations [61].

#### 3.2.2. Ionizing Radiation

With each exposure to ionizing radiation, there is a risk of possible cellular and DNA lesions, which, if not repaired, may lead to more or less serious consequences to cellular functions and manifest themselves in the individual after variable times depending on the amount of radiation absorbed and the manner in which the radiation itself was administered [62].

The reduction of risks associated with medical exposures to ionizing radiation prescribed in Legislative Decree 101/2020 (previously in Legislative Decree 187/2000), according to an article reported by INAIL, rests on the prohibition of unjustified exposures and on the principle of optimization, whereby all doses from medical exposures, excluding radiotherapy, must be kept at the lowest achievable level. This principle of optimization relates in particular to the choice of equipment, appropriateness with respect to the diagnostic information or therapeutic result to be achieved, quality control, examination and evaluation of the doses or activities administered to the patient, and obtaining the same diagnostic information with techniques without the use of ionizing radiation or with a lower radiation dose [63,64,65].

### 3.3. Biological Pollutants

The concept of biohazards is linked to a series of characteristics: infectivity, transmissibility, pathogenicity, neutralizability, source, reservoir, and dose.

According to Legislative Decree 81/08, microorganisms are divided into four hazard classes with increasing values from one to four, depending on the hazard factors (Table 2) [66].

According to the Document for the Evaluation of Risks to Workers’ Health and Safety (Legislative Decree No. 81 of 9 April 2008), reported by the National Association of Italian Dentists (ANDI), health professionals are potentially exposed to the risk of infection as a result of possible contact with infected biological materials [67].

The following are necessary for preventive purposes: risk assessment, adequate training of personnel, implementation of preventive and protective interventions following the correct procedure [68].

In addition to indoor pollutants, which indirectly harm the outdoor environment, it is worth noting the presence of outdoor pollutants.

A significant amount of carbon emissions from dentistry come from patients’ car journeys and the transport of materials, which affect air quality, releasing over 443 tons of nitrogen oxide (NOx) and 22 tons of particulate matter (PM2.5) per year [69].

Using advances in information technology, the number of physical appointments can be reduced, and patients should be encouraged to walk or cycle. For example, King’s College London offered incentives such as pedometers and guided walks to staff. This would improve people’s health, save fuel and parking costs, and reduce carbon emissions [70].

A further problem lies in the high consumption of electricity for electronic equipment, a high water consumption, the use of radiation, and the production of waste such as plastic as well as hazardous waste that includes harmful metals and potentially toxic products [71].

Green dentistry concept

The concept of ’green dentistry’ first appeared at the Fifth Congress of the European Association of Dental Students in Belgrade, Serbia, in March 2003, when the Greek delegation defined the guidelines and proposed the adoption of the project by the assembly. Currently, the countries that have adopted this project are Croatia, Sweden, the Netherlands, the United Kingdom, and Greece [72]. Its main domains are:○An increased awareness and environmental sensitivity among dental professionals;○The encouragement of procedures/regulations/policies compatible with the EU’s sustainable development strategy;○The establishment of a network for cooperation, information exchange, and opinions on eco-friendly dental practice in Europe and worldwide [73].

Green dentistry encompasses four categories of green practice and provides an ecological model for dental clinics [74,75]:○Waste reduction;○Pollution prevention;○The minimization of water and energy resource usage;○High-tech dentistry.

### 3.4. Waste Reduction

According to the article “Ecofriendly Dentistry and Green Hospitals” by Sodhi et al., published in the Journal of Advanced Medical and Dental Sciences Research in May 2019, the fronts that need to be addressed to reduce pollution in dental practices are as follows (Table 3) [76]:

### 3.5. Pollution Prevention

According to the Webster dictionary, pollution is defined as the contamination of the air, soil, or water due to the discharge of harmful substances. The Council on Dental Practice of the American Dental Association has outlined a list to implement eco-friendly and environmentally friendly practices in dental clinics [77]:○The installation of an amalgam separator and ensuring that amalgam waste is properly recycled; the use of HVE (High Volume Evacuation) and water spray to reduce dust and vapor levels during the removal of amalgam restorations;○The elimination of loose mercury use and training staff on the proper handling, management, and disposal of Hg-containing materials, including previously restored amalgam dental elements;○The avoidance of using bleach (sodium hypochlorite) to disinfect aspirator tubes (e.g., after surgery), as it accelerates mercury release from amalgam;○Mercury waste amalgam should be stored in an appropriate sealed container; amalgam separators compliant with International Organization for Standardization (ISO) Standard 11143, if used with capture devices and vacuum pump filters, can achieve a removal efficiency of over 95%;○The utilization of digital X-rays (digital imaging) instead of conventional systems: this eliminates toxic X-ray fixer solutions and lead foils. Additionally, it has other advantages such as the immediate availability of images, better quality, higher diagnostic efficacy, and minimal radiation exposure; dental suppliers should be instructed to reduce packaging and bulk in order to minimize shipping waste;○Recycling old and damaged manual instruments by giving them a new life;○The use of biodegradable and approved surface disinfectants and cleaners; the use of reusable stainless steel or compostable impression trays;○Water lines should be regularly disinfected using biodegradable or enzymatic cleaners. Chlorine bleach, however, should not be used for cleaning, as it can release mercury into the clinic air. All water consumed within the clinic should be filtered, as it decreases calcium and other deposits, increases tool longevity, and reduces maintenance needs; steam sterilization should be utilized—an efficient procedure in terms of time and reliability, it removes toxic sterilization chemical vapors from the environment and eliminates hazardous waste [78];○The elimination of chemicals such as glutaraldehyde, a potent skin irritant, should be reduced, as it is toxic; authorized handlers should always be used for the off-site recycling of hazardous materials; reusable and autoclavable stainless steel rinse cups should be utilized for patient rinsing instead of disposable plastic or paper cups; the use of monomers should be reduced or eliminated; all disposable items should be replaced with reusable materials whenever possible [79];○According to authors Sodhi et al., one of the first things a green hospital or medical dental office should do before implementing a sustainability program is to create a team to organize and monitor this initiative [80];○The team should be authorized and supported by leadership and motivated towards specific goals. Document shredding services: most patient paper documents are thrown away in bins for recycling, which could jeopardize their privacy. It is therefore important to hire a provider who collects and securely shreds them according to HIPAA regulatory standards;○The retreatment of medical devices: the proper retreatment and sterilization of medical devices such as gowns, gloves, and towels helps reduce the amount of waste entering landfills;○Document-management services: streamlining printing and copying can reduce costs and improve productivity, such as converting documentation to computerized records and sending email and text messages as appointment reminders instead of written notes; the reselling and recycling of excess equipment can generate revenue;○LEED certification. Having a LEED (Leadership in Energy and Environmental Design)-certified building will help lower operating costs, conserve energy, water, and other resources, provide a healthier environment for occupants, and qualify for savings incentives such as tax breaks.

### 3.6. High-Tech Dentistry

Constantly improving technologies could promote the concept of green dentistry, in particular, through the following developments:○Digital X-rays;○CAD/CAM systems;○Highly aesthetic prosthetic materials that have replaced amalgam;○Equipment for the disposing of sharp objects that renders them inert;○Steam sterilizers that eliminate the use of harmful chemicals;○The digitization of documentation (medical records) and communication with patients (memos, e-email, websites as the main marketing tools);○Diode lasers, which eliminated the need for shrink-wrap cables;○Oil-free compressors [81].

The advantages of a digital workflow include the ability to streamline processes that can be complicated by the analogue method, the reduction of chair time, the simplification and standardization of laboratory procedures, and the quality and precision of manufactured products using CAD/CAM technology [81,82].

4R rule in healthcare and dentistry

One of the simplest ways to initiate a green initiative is to develop a waste-reduction plan. According to the ministerial decree issued by the Ministry of Health on 5 February 1998, biomedical waste is defined as “any waste generated during the diagnosis, management, or immunization of human or animal subjects, during research activities associated with biological materials”. These regulations have made it mandatory for healthcare facilities to implement management systems to segregate, disinfect, and dispose of all waste in an eco-friendly manner, as there is a high risk of nosocomial infections among patients and healthcare workers due to poor waste management. Therefore, the best disposal options should be used to prevent or reduce the release of hazardous substances from dental clinics into the environment. The World Bank’s healthcare guidelines have provided a four-step list for managing healthcare waste:Segregation;Transportation;Treatment;Final disposal.

Producers of biomedical waste should therefore opt for universal precautions and appropriate safety instructions during therapeutic and diagnostic procedures. They should use non-chlorinated plastic bags for the disposal of incinerable waste. According to Pockrass et al., green dentistry is based on the “Four Rs” model: Rethink, Reduce, Reuse, and Recycle. The Four Rs is a strategy implemented by dental professionals to facilitate the transition to a more sustainable dental practice [83,84]. The implementation of small and convenient changes can have a significant impact on long-term environmental sustainability.

Rethink: Environmentalism and sustainability are both considered mindsets, and every decision is made with a certain mentality. Therefore, reshaping a mindset is the first strategy for change. Dental practices should rethink how they are managed by implementing simple changes such as reducing energy and water consumption [85].

Reduce: “Reduce” means minimizing the quantity of materials by reusing and recycling as much as possible in clinical practice [86]. To alleviate pressure on Earth’s resources, healthcare providers should reduce their consumption of resources. For example, to prevent deforestation and slow global warming, we should reduce our paper consumption and consequently waste production. Having a dental practice that reduces or eliminates paper usage involves transitioning to a digital system, such as converting patient files, medical histories, and other documentation into an electronic system. The elimination of paper not only makes information sharing easier and more accessible but also protects personal information. This saves money, increases productivity, saves space as no archive is needed, and is a great way to ensure that medical records are more accurate. Also, using digital radiography keeps all patient records in one place, reduces radiation exposure, and allows clinical images and photographs to be shared without losing quality. Resource reduction can occur in several ways:1.Water conservation:○Follow hand hygiene guidelines from the Centers for Disease Control and Prevention and use hand sanitizer instead of washing hands when not visibly soiled;○When handwashing is necessary, turn off the water while soaping;○Participate in the “Save 90 A Day” campaign to educate patients to turn off the water while brushing their teeth;○Only run full loads in the autoclave. Low-flow aerators can be installed on all sink faucets;○Check for water leaks throughout the practice every six months.
2.Reduce the consumption of disposable items. Eliminating paper usage means using computers and digital technology whenever possible to create, use, and store office documents. Eliminate the use of plastic bags and packaging by replacing them with paper when possible.


Reuse: To save resources and energy, this practice encourages the use of reusable and biodegradable items:○Reusable cotton towels for the operating room instead of disposable plastic or paper bibs;○High- and low-volume stainless steel aspirator tips for surgical/endodontic suction, as an alternative to disposable plastic;○Reusable glass irrigation syringes;○Biodegradable or autoclavable disposable cups;○Recycled chlorine-free paper products instead of traditional paper products.

Recycle: Most waste material labeled as waste can be processed into a new product. Many materials such as paper/plastic cups, paper, magazines, napkins, and infection-control waste can be recycled, thus reducing overall costs. The phases outlined by the recycling triangle are represented by three green arrows. The first arrow symbolizes the collection of materials to be recycled. “Regeneration”, i.e., the creation of something new with recycled materials, is depicted by the second arrow, and “resale”, i.e., offering for sale items made with recycled materials, is illustrated by the third. To “recycle” dental materials, specific existing programs need to be identified. Taking note of what could be improved in terms of environmental respect is an effective way to incorporate “Going green” into everyday life [86].

## 4. Discussion

The results of this study highlight the importance of adopting green dentistry practices to mitigate the environmental impact of dental procedures. Several authors have examined various aspects of pollution related to dental practices, and the comparisons between these studies provide an overview of the challenges and possible solutions.

Regarding the past use of dental amalgam containing mercury, Tibau and Grube (2019) highlighted how this substance, once released into the environment, can bioaccumulate in organisms, posing a significant risk to human health and the ecosystem [12]. Similarly, Dudeja et al. (2023) pointed out that mercury in dental amalgam can cause neurotoxicity and other serious health conditions, and argued for the need to phase out the use of this material [23].

By contrast, Worthington et al. (2021) reported that resin composites, considered an alternative to mercury, may have a shorter lifespan than traditional amalgams, raising doubts about their long-term effectiveness and sustainability. However, reducing the use of mercury remains a priority to reduce the overall environmental impact [25].

Concerning the use of solvents in dentistry, Rajan et al. (2019) showed that prolonged exposure to xylene vapors can cause respiratory tract irritation and damage to the central nervous system [39].

Similarly, Vajrabhaya et al. (2004) pointed out that chloroform, once used as a solvent, is now recognized for its toxic and carcinogenic effects [42].

On the other hand, Martos et al. (2011) proposed the use of solvents of natural origin such as eucalyptus oil and limonene, which are less harmful than traditional solvents while maintaining comparable efficacy. This suggests that the adoption of natural alternatives may be a step towards more sustainable and safer practices [38].

A further topic under study was the use of disinfectants. Beauchamp et al. (1992) pointed out that glutaraldehyde, although effective as a disinfectant, can be highly toxic and irritating to the skin and respiratory tract [46]. Furthermore, Saccucci et al. (2017) indicated that the use of formaldehyde, although a potent disinfectant, poses carcinogenic risks to exposed healthcare workers [51].

By contrast, Sebastiani et al. (2017) suggested the use of less-toxic disinfectants, such as peracetic acid at low concentrations, which significantly reduce health risks without compromising disinfection effectiveness. These studies highlight the need to balance disinfection efficacy with safety for operators and the environment [52].

In order to safeguard patients, operators, and the environment, the concept of green dentistry has gained significant importance over the years.

Landrigan et al. (2020) highlighted the ‘Four Rs’ model as a clear guide to improving sustainability. Their systematic approach to rethink, reduce, reuse, and recycle is well-founded in the context of environmental management. However, while the model provides a comprehensive framework, practical challenges may arise in implementing all four steps, especially in dental clinics with limited resources [84].

Duane et al. (2019) emphasized the critical importance of staff training and environmental education as fundamental to the successful adoption of green dentistry practices. Their emphasis on cultural change within dental clinics is essential to ensure that new sustainable practices are permanently integrated into daily operations. However, it may be necessary to balance investment in training with other upfront costs for the implementation of sustainable technologies and materials [70].

Vandenberghe (2020) proposed digitization as a key strategy to reduce the environmental impact of dental clinics through the adoption of digital X-rays and electronic medical record management. This approach not only reduces the use of resources such as paper and chemicals but also improves overall operational efficiency. However, digitization may face resistance from healthcare professionals who are less inclined to technology or concerned about data security [82].

Sodhi et al. (2019) emphasized the use of reusable materials as an effective way to reduce waste generated by dental clinics. The adoption of reusable stainless steel instruments and sustainable materials offers significant environmental benefits and can also reduce long-term costs. However, the adoption of these materials may be limited by economic and logistical considerations as well as local regulations [80].

Cheema (2019) promoted the implementation of recycling programs to manage the ecological footprint of dental clinics. Recycling materials such as metal, plastic, and paper not only reduces resource consumption but can also enhance the reputation of clinics as socially responsible entities. However, the availability and accessibility of recycling infrastructure can vary significantly between regions, affecting the effectiveness of such initiatives [69].

## 5. Conclusions

Within the limitations of the present study, the concept of green dentistry could be applied effectively by ensuring that the measures already implemented to reduce indoor and outdoor risk factors are sustained and enhanced. This entails ongoing efforts to mitigate the environmental impacts associated with dental practices while simultaneously improving the overall sustainability of the profession.

The continued implementation of measures aimed at reducing indoor risk factors involves maintaining proper ventilation systems to minimize the concentration of airborne contaminants, such as aerosols generated during dental procedures. Utilizing high-efficiency particulate air (HEPA) filters and incorporating adequate air exchange rates in dental offices can help mitigate indoor air pollution and promote a healthier environment for both patients and staff.

Furthermore, it is essential to continue efforts to minimize outdoor risk factors by responsibly managing waste materials generated in dental practices. This includes the proper disposal and recycling of materials such as dental amalgam, which contains mercury and can pose environmental risks if not handled correctly. Implementing effective waste management practices, such as using amalgam separators and ensuring the proper disposal of hazardous materials, is crucial for reducing the environmental footprint of dental clinics.

In addition to these measures, ongoing improvements in technology and practices can further enhance the sustainability of dental care. This includes embracing digital dentistry solutions, such as digital radiography and CAD/CAM systems, which reduce material waste and energy consumption compared to traditional methods. Investing in energy-efficient equipment and adopting environmentally friendly materials can also contribute to the greening of dental practices.

Overall, the concept of green dentistry remains applicable and achievable, provided that efforts to reduce indoor and outdoor risk factors are continued and improved upon. By incorporating sustainable practices into everyday operations and embracing technological advancements, dental professionals can contribute to a healthier environment and a more sustainable future for dental care.

## Figures and Tables

**Table 1 dentistry-13-00038-t001:** Inclusion and exclusion criteria.

Criteria	Inclusion	Exclusion
Study Type	Original research articles, i.e., clinical trials, randomized clinical trials, observational studies, reviews, meta-analyses, and systematic reviews	Non-full-text articles, editorials, and commentaries
Language	English	Studies in languages other than English
Subject	Human studies relevant to the impact of dental practices on the environment or green dentistry	Animal studies or in vitro studies
Focus	Studies assessing pollutants, eco-friendly materials, and energy-efficient technologies in dentistry	Studies not related to environmental impact or sustainability in dentistry
Publication Date	No filters were added concerning the publication date.	

**Table 2 dentistry-13-00038-t002:** Pathogen hazard classes.

Group	Description	Examples of Agents
1	None or low individual and collective risk	Agents with a low probability of causing diseases in humans or animals.
		Unspecified agents with low pathogenic potential.
2	Moderate individual risk, limited collective risk	Pathogenic agents that can cause diseases in humans or animals, but do not pose a serious danger to the community or the environment.
		Salmonella paratyphi A, B, C Mumps virus Poliovirus
3	High individual risk, low collective risk	Biological agents that can cause severe diseases in humans and pose a serious risk to workers, but effective prophylactic or therapeutic measures are usually available.
		Mycobacterium tuberculosis Salmonella typhi Hepatitis B virus Hepatitis C virus Human immunodeficiency virus (HIV)
4	High individual and collective risk	Biological agents that can cause severe diseases in humans, pose a serious risk to workers, and may also present a high risk of spreading within the community.
		COVID-19 Crimean–Congo hemorrhagic fever virus Ebola virus

**Table 3 dentistry-13-00038-t003:** Alternatives and recommendations to mitigate the environmental and health risks associated with conventional dental procedures.

Procedure	Issues	Alternatives/Recommendations
Infection Control Procedures	○ Endangering the health of healthcare workers; ○ Contributing to poor air quality in operational areas; ○ Polluting the community water flow.	Non-toxic alternatives are more effective in protecting the health and safety of healthcare workers, patients, and society as a whole.
Conventional X-ray Systems	○ Producing toxic waste that dental practices must dispose of, often into public sewage systems; ○ Chemical fixatives and lead foils used in conventional X-ray processes contain harmful substances; ○ Lead-containing foils are a neurotoxin and can persist in the environment for up to 2000 years if not disposed of correctly.	Utilize alternatives such as digital X-ray systems that do not produce toxic waste.
Conventional Dental Aspiration Systems	A high water consumption, contributing to water scarcity during a global water crisis.	Consider high-tech dry aspiration systems as alternatives that achieve efficient suction results without using water.
Handling and Removal of Dental Material Containing Mercury	○ Dental amalgam waste releases mercury into the environment; ○ The direct and indirect exposure of dental staff to mercury emissions; ○ Improper handling of amalgam waste can lead to environmental pollution.	Implement strict regulations for handling amalgam and its waste to reduce environmental pollution. Utilize the “Four Rs” rule (Rethink, Reduce, Reuse, Recycle) to limit the environmental impact of dental waste [76].

## Data Availability

No new data were created or analyzed in this study. Data sharing is not applicable to this article.

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
