# Peer review of "Green Dentistry: State of the Art and Possible Development Proposals"

_dentistry, 2025, doi:10.3390/dj13010038_

Round 1
Reviewer 1 Report
Comments and Suggestions for Authors
Dear Authors,
Thank you very much for this paper which highlights the importance of sustainability and how to incorporate that in the dental field. The manuscript is well written and well organised.
below are my comments
Materials and methods,
regarding the literature search, when was the start and end dates ?
line 684:
patent should be removed.
Author Response
REVIEWER 1
Dear Reviewer,
we thank you for your valuable suggestions,
in accordance with which we have proceeded to make the following changes.
We hope to have improved the paper to your satisfaction and thank you for your kindness and helpfulness while remaining available for further clarification.
Dear Authors,
Thank you very much for this paper which highlights the importance of sustainability and how to incorporate that in the dental field. The manuscript is well written and well organised.
below are my comments
Materials and methods,
regarding the literature search, when was the start and end dates?
We added the literature search period as follows:
From September 2023 to September 2024, an extensive search of electronic databases, including Cochrane Oral Health Group Specialized Register, Cochrane Central Register of Controlled Trials (CENTRAL), Web of Science, PubMed, EMBASE and Google Scholar, was performed.
line 684:
patent should be removed.
It was removed.
Reviewer 2 Report
Comments and Suggestions for Authors
This review highlights all dental procedures attributable to sectoral waste and also includes possible alternatives in line with the concept of sustainable development, concluding that green dentistry may be applicable whether measures already taken to reduce indoor and outdoor risk factors are continued and improved.
INTRODUCTION
Line 43 please correct English using “can only be partially achieved”.
Overall introduction does not analyze “sustainable development” concepts or its principles; I suggest adding more information about development models and characteristics. Also, I suggest adding the association with dental procedures and sectoral waste.
MATERIALS AND METHODS
Materials and methods does not report any diagram, I suggest adding a decision flow chart or a table with inclusion and exclusion criteria. Also, I suggest reporting the studied characteristics.
RESULTS
Results are well-described and organized.
DISCUSSION
Discussion is well-organized and also well-describes the included characteristics.
CONCLUSION
Conclusions are in line with the aims and the discussion of this study. Also, I appreciate the future directions reported.
Author Response
Dear Reviewer,
we thank you for your valuable suggestions,
in accordance with which we have proceeded to make the following changes.
We hope to have improved the paper to your satisfaction and thank you for your kindness and helpfulness while remaining available for further clarification.
This review highlights all dental procedures attributable to sectoral waste and also includes possible alternatives in line with the concept of sustainable development, concluding that green dentistry may be applicable whether measures already taken to reduce indoor and outdoor risk factors are continued and improved.
INTRODUCTION
Line 43 please correct English using “can only be partially achieved”.
The phrase was modified as follows:
Consequently, without a healthy environment, public health, economic stability, and social stability can only be partially achieved [3].
Overall introduction does not analyze “sustainable development” concepts or its principles; I suggest adding more information about development models and characteristics. Also, I suggest adding the association with dental procedures and sectoral waste.
According to your suggestions, the introduction was modified adding several informations as follows:
Sustainable development is a multidimensional concept aimed at meeting the needs of present generations without compromising the ability of future generations to meet their own needs. It encompasses three interrelated pillars: economic, environmental, and social sustainability. Economic sustainability emphasizes growth models that are inclusive, equitable, and maintain resources for long-term prosperity. Environmental sustainability aims to conserve biodiversity, reduce pollution, and ensure the responsible use of natural resources. Social sustainability focuses on creating systems that improve living standards, ensure fair access to resources, and promote equity and justice within communities [6,7].
In recent decades, development models have evolved to incorporate principles of sustainability, transitioning from resource-intensive, linear approaches to circular models that prioritize resource efficiency, waste reduction, and ecosystem health. Key characteristics of these models include renewable energy adoption, eco-design principles, and the integration of life-cycle assessments into planning and decision-making processes. Such frameworks align with the global agenda for sustainable development, as outlined in international agreements like the United Nations Sustainable Development Goals (SDGs) [8].
The principles of sustainable development are particularly relevant in the healthcare and dental sectors, where procedures generate significant amounts of waste and consume considerable resources. The dental industry, in particular, contributes to sectoral waste through the use of disposable materials, packaging, and chemical substances that pose challenges for waste management and environmental health. Addressing these issues involves rethinking the materials and methods used in dental practices to reduce ecological footprints and promote sustainability. Initiatives such as the adoption of biodegradable materials, recycling programs, and energy-efficient technologies are steps toward aligning dental procedures with sustainable development goals.
MATERIALS AND METHODS
Materials and methods does not report any diagram, I suggest adding a decision flow chart or a table with inclusion and exclusion criteria. Also, I suggest reporting the studied characteristics.
The informations required were added as follows:
The table below summarizes the criteria used for study selection (Table 1).
Table 1. Inclusion and exclusion criteria.
Criteria Inclusion Exclusion
Study Type Original research articles i.e. clinical trial, randomized clinical trial, observational studies, reviews, meta-analyses, and systematic reviews Non-full-text articles, editorials, and commentaries
Language English Studies in languages other than English
Subject Human studies relevant to the impact of dental practices on the environment or green dentistry Animal studies or in vitro studies
Focus Studies assessing pollutants, eco-friendly materials, energy-efficient technologies in dentistry Studies not related to environmental impact or sustainability in dentistry
Publication Date No filters were added concerning publication date.
The decision-making process for study selection is illustrated in the following flow chart:
1. Identification: Database search yielded a total of 1,230 studies. After removing duplicates, 890 unique studies remained.
2. Screening: Titles and abstracts were screened, resulting in the exclusion of 720 irrelevant studies.
3. Eligibility: Full texts of 170 studies were assessed against the inclusion and exclusion criteria.
4. Inclusion: A final selection of 76 studies was included for qualitative analysis.
The included studies analyzed the following key characteristics:
1. Pollutants in Dentistry: Emissions, waste management, and use of non-biodegradable materials.
2. Eco-friendly Alternatives: Adoption of biodegradable materials, non-toxic disinfectants, and sustainable packaging.
3. Energy Efficiency: Integration of energy-saving devices and technologies in dental practices.
4. High-Tech Dentistry: Digital tools and systems reducing material waste and carbon footprints.
5. Compliance and Awareness: Levels of compliance with environmental regulations and awareness among dental professionals about green practices.
RESULTS
Results are well-described and organized.
DISCUSSION
Discussion is well-organized and also well-describes the included characteristics.
CONCLUSION
Conclusions are in line with the aims and the discussion of this study. Also, I appreciate the future directions reported.
Reviewer 3 Report
Comments and Suggestions for Authors
Thank you for giving me an opportunity to review this manuscript.
This is an interesting topic. I have few comments which I believe will be helpful in improving the quality of this paper.
11. To assure the quality of narrative reviews, they should follow some guidelines and the authors should provide a checklist. Please confirm if this narrative review follows any guidelines (“SANRA” etc. ) and cite the same….or there is any checklist available for this review (please provide the same)
22. Introduction: A. Please add some previously published work in the introduction. B. Introduce how dental materials etc. are causing environmental hazard.
33. Methodology:
A. Please elaborate this section.
B. Provide search details in main file or as supplementary file.
C. Provide details about when this search was done, who all performed the search. how selection bias was removed
4. Results and conclusion sections are well written
5. After conclusion section, please check this term “6. patents”
Author Response
Dear Reviewer,
we thank you for your valuable suggestions,
in accordance with which we have proceeded to make the following changes.
We hope to have improved the paper to your satisfaction and thank you for your kindness and helpfulness while remaining available for further clarification.
Thank you for giving me an opportunity to review this manuscript.
This is an interesting topic. I have few comments which I believe will be helpful in improving the quality of this paper.
11. To assure the quality of narrative reviews, they should follow some guidelines and the authors should provide a checklist. Please confirm if this narrative review follows any guidelines (“SANRA” etc. ) and cite the same….or there is any checklist available for this review (please provide the same)
The following sentence was added according to the followed scale used for study realization.
The SANRA scale was applied to assess study validity (https://rossisanusi.wordpress.com/wp-content/uploads/2022/03/sanra.pdf).
22. Introduction: A. Please add some previously published work in the introduction. B. Introduce how dental materials etc. are causing environmental hazard.
According to your suggestions, the introduction was modified adding several informations as follows:
Sustainable development is a multidimensional concept aimed at meeting the needs of present generations without compromising the ability of future generations to meet their own needs. It encompasses three interrelated pillars: economic, environmental, and social sustainability. Economic sustainability emphasizes growth models that are inclusive, equitable, and maintain resources for long-term prosperity. Environmental sustainability aims to conserve biodiversity, reduce pollution, and ensure the responsible use of natural resources. Social sustainability focuses on creating systems that improve living standards, ensure fair access to resources, and promote equity and justice within communities [6,7].
In recent decades, development models have evolved to incorporate principles of sustainability, transitioning from resource-intensive, linear approaches to circular models that prioritize resource efficiency, waste reduction, and ecosystem health. Key characteristics of these models include renewable energy adoption, eco-design principles, and the integration of life-cycle assessments into planning and decision-making processes. Such frameworks align with the global agenda for sustainable development, as outlined in international agreements like the United Nations Sustainable Development Goals (SDGs) [8].
The principles of sustainable development are particularly relevant in the healthcare and dental sectors, where procedures generate significant amounts of waste and consume considerable resources. The dental industry, in particular, contributes to sectoral waste through the use of disposable materials, packaging, and chemical substances that pose challenges for waste management and environmental health. Addressing these issues involves rethinking the materials and methods used in dental practices to reduce ecological footprints and promote sustainability. Initiatives such as the adoption of biodegradable materials, recycling programs, and energy-efficient technologies are steps toward aligning dental procedures with sustainable development goals.
33. Methodology:
A. Please elaborate this section.
B. Provide search details in main file or as supplementary file.
C. Provide details about when this search was done, who all performed the search. how selection bias was removed
The informations required were added as follows:
The table below summarizes the criteria used for study selection (Table 1).
Table 1. Inclusion and exclusion criteria.
Criteria Inclusion Exclusion
Study Type Original research articles i.e. clinical trial, randomized clinical trial, observational studies, reviews, meta-analyses, and systematic reviews Non-full-text articles, editorials, and commentaries
Language English Studies in languages other than English
Subject Human studies relevant to the impact of dental practices on the environment or green dentistry Animal studies or in vitro studies
Focus Studies assessing pollutants, eco-friendly materials, energy-efficient technologies in dentistry Studies not related to environmental impact or sustainability in dentistry
Publication Date No filters were added concerning publication date.
The decision-making process for study selection is illustrated in the following flow chart:
5. Identification: Database search yielded a total of 1,230 studies. After removing duplicates, 890 unique studies remained.
6. Screening: Titles and abstracts were screened, resulting in the exclusion of 720 irrelevant studies.
7. Eligibility: Full texts of 170 studies were assessed against the inclusion and exclusion criteria.
8. Inclusion: A final selection of 76 studies was included for qualitative analysis.
The included studies analyzed the following key characteristics:
6. Pollutants in Dentistry: Emissions, waste management, and use of non-biodegradable materials.
7. Eco-friendly Alternatives: Adoption of biodegradable materials, non-toxic disinfectants, and sustainable packaging.
8. Energy Efficiency: Integration of energy-saving devices and technologies in dental practices.
9. High-Tech Dentistry: Digital tools and systems reducing material waste and carbon footprints.
10. Compliance and Awareness: Levels of compliance with environmental regulations and awareness among dental professionals about green practices.
4. Results and conclusion sections are well written
5. After conclusion section, please check this term “6. patents”
The term was removed.
Reviewer 4 Report
Comments and Suggestions for Authors
The presented article has a very intriguing and interesting topic.
Taking up the problem of green dentistry in an era of decreasing resources of materials such as cobalt and titanium requires the introduction of a sustainable method of its use.
Unfortunately, this article only slightly skims the topic. In fact, it is off-topic. pages 2-9 have nothing to do with the concept of green dentistry described on the following pages.
At this point, one should also strongly argue with the statements that it is better to use reusable materials. This is clearly untrue. We should use disposable but recyclable materials in dental treatment. The use of reusable tubes for suction devices or other listed elements does not increase patient safety.
The authors have completely superficially skimmed the topic of the production of prosthetic restorations, writing only that CAD/CAM is able to reduce the consumption of materials.
The article is poorly prepared. The descriptions of the use of amalgam or other solvents in the description concern only their toxicity and not the reduction of their use.
Green dentistry is about reducing the use of materials, using optimal procedures and being responsible for the planet. It does not include in any way whether something is toxic or non-toxic to the Patient, and that is what most of the description is about.
In my opinion, the authors do not understand the subject and introduce chaos into the presented problem.
In my opinion, the article cannot be improved and should be rejected.
Author Response
Dear Reviewer,
Thank you for your detailed and constructive feedback on our manuscript. We greatly appreciate the time and effort you have taken to provide us with your insights, which will guide us in refining our work.
We acknowledge your concerns regarding the focus and depth of our manuscript. Your observation that certain sections, particularly pages 2–9, may deviate from the main topic of green dentistry is noted, and we regret any lack of clarity or coherence in addressing the core principles of sustainable dental practices. We will ensure that the content is aligned with the concept of green dentistry, focusing on reducing material usage, optimizing procedures, and emphasizing environmental responsibility.
We also understand your concerns about the discussion of reusable materials and their safety compared to disposable but recyclable alternatives. Our intention was to explore a wide range of approaches to sustainability in dentistry, but we recognize the importance of highlighting the practicality and effectiveness of recyclable disposables in modern dental practices.
Regarding the discussion of CAD/CAM technologies, we acknowledge that the coverage may have been superficial and lacked detail on their potential to reduce material consumption.
Thank you again for your thoughtful review and for helping us to better address this critical and timely topic.
We remain at your disposal for any further clarification.
Taking up the problem of green dentistry in an era of decreasing resources of materials such as cobalt and titanium requires the introduction of a sustainable method of its use.
Unfortunately, this article only slightly skims the topic. In fact, it is off-topic. pages 2-9 have nothing to do with the concept of green dentistry described on the following pages.
At this point, one should also strongly argue with the statements that it is better to use reusable materials. This is clearly untrue. We should use disposable but recyclable materials in dental treatment. The use of reusable tubes for suction devices or other listed elements does not increase patient safety.
The authors have completely superficially skimmed the topic of the production of prosthetic restorations, writing only that CAD/CAM is able to reduce the consumption of materials.
The article is poorly prepared. The descriptions of the use of amalgam or other solvents in the description concern only their toxicity and not the reduction of their use.
Green dentistry is about reducing the use of materials, using optimal procedures and being responsible for the planet. It does not include in any way whether something is toxic or non-toxic to the Patient, and that is what most of the description is about.
In my opinion, the authors do not understand the subject and introduce chaos into the presented problem.
In my opinion, the article cannot be improved and should be rejected.
Round 2
Reviewer 3 Report
Comments and Suggestions for Authors
authors have performed the changes as requested. I believe the manuscript can be published in the present form.